

# Erector spinae plane block versus paravertebral block on postoperative quality of recovery in obese patients undergoing laparoscopic sleeve gastrectomy: a randomized controlled trial

Guanyu Yang, Pengfei Wang, Yue Yin, Huan Qu, Xin Zhao, Xiaogao Jin and Qinjun Chu

Department of Anesthesiology and Perioperative Medicine, Zhengzhou Central Hospital Affiliated to Zhengzhou University, Zhengzhou, Henan, China

## ABSTRACT

**Purpose.** To compare the impact of erector spinae plane block (ESPB) and paravertebral block (PVB) on the quality of postoperative recovery (QoR) of patients following laparoscopic sleeve gastrectomy (LSG).

**Methods.** A total of 110 patients who underwent elective LSG under general anesthesia were randomly assigned to receive either ultrasound-guided bilateral ESPB or PVB at T8 levels. Before anesthesia induction, 40 mL of 0.33% ropivacaine was administered. The primary outcome was the QoR-15 score at 24 hours postoperatively.

**Results.** At 24 hours postoperatively, the QoR-15 score was comparable between the ESPB and PVB groups (131 (112–140) *vs.* 124 (111–142.5), $P = 0.525$). Consistently, there was no significant difference in QoR-15 scores at 48 hours postoperatively, numerical rating scale (NRS) pain scores at any postoperative time points, time to first ambulation, time to first anal exhaust, postoperative cumulative oxycodone consumption, and incidence of postoperative nausea and vomiting (PONV) between the two groups (all $P > 0.05$). No nerve block-related complications were observed in either group.

**Conclusion.** In patients undergoing LSG, preoperative bilateral ultrasound-guided ESPB yields comparable postoperative recovery to preoperative bilateral ultrasound-guided PVB.

Corresponding author
Qinjun Chu, jimmynetchu@163.com

## INTRODUCTION

Obesity is a prevalent medical condition that has often been associated with hypertension, dyslipidemia, cardiovascular disease, and type 2 diabetes mellitus (*Smith & Smith, 2016*; *Varban et al., 2017*). Surgical interventions, such as laparoscopic sleeve gastrectomy (LSG), are commonly used for long-term weight management in obese individuals (*Glass et al., 2019*; *Alharbi, 2020*). However, many patients who undergo bariatric surgery experience

significant acute postoperative pain, making adequate postoperative pain management crucial for their recovery (*Nimmo, Foo & Paterson, 2017*). Multimodal analgesia, including regional analgesia, has become the standard of care for managing acute pain. In recent years, the use of regional blocks in multimodal analgesia has experienced a significant rise.

Paravertebral block (PVB) and erector spinae plane block (ESPB) are currently widely used nerve block methods in clinical practice. While the analgesic effects of ESPB and PVB have been well-established in postoperative analgesia for thoracic surgery (*Yao et al., 2020*; *Taketa, Irisawa & Fujitani, 2020*; *Koo et al., 2022*; *D'Ercole, Arora & Kumar, 2018*), there is limited research on the analgesic effects of these methods after upper abdominal surgery, particularly in patients with obesity.

Therefore, we conducted this trial to compare the effects of these two analgesic methods, PVB and ESPB, on the postoperative quality of recovery (QoR) and pain in patients undergoing LSG. It was hypothesized that there would be no difference in the effectiveness of the two methods.

## MATERIALS & METHODS

This randomized controlled clinical study was initiated after obtaining approval from the Medical Ethics Committee of Zhengzhou Central Hospital (ethical number: 202109, date: 21/1/2021). The study was registered on http://www.chictr.org.cn on 30/1/2021 (ChiCTR2100042846) and was conducted in accordance with the Helsinki Declaration.

This study included patients who underwent surgery at Zhengzhou Central Hospital between February 2021 and June 2021. Written informed consent was obtained from patients after a detailed explanation of the trial. The inclusion criteria were as follows: (1) Patients scheduled for LSG; (2) patients aged 18–45 years; (3) body mass index (BMI) between 30 kg/m$^2$ and 40 kg/m$^2$; (4) American Society of Anesthesiologists (ASA) physical status II or III. The exclusion criteria were: (1) hematological diseases or blood coagulation dysfunction; (2) infection at the puncture site; (3) allergy to local anesthetics or nonsteroidal antiinflammatory drugs (NSAIDs); (4) history of psychiatric or neurological diseases; (5) inability to communicate effectively.

### Study protocol

Patients were randomly allocated into the ultrasound-guided ESPB group (Group ESPB) or ultrasound-guided PVB group (Group PVB) at T8 levels using random numbers generated by Excel. The researchers got ready sealed, opaque, consecutively numbered envelopes holding cards with assignment details from the list. When eligible patients arrived in the operating room, the investigator nurse allocated participants by opening the envelopes with the pre-written allocations. The allocation to groups was unaware to patients, surgeons, post-anesthesia care unit (PACU) staff, data collectors, and those performing statistical analyses throughout the observation period, including all post-operative follow-up periods. All patients received general anesthesia after regional blocks were performed by the same anesthesiologist with more than 3 years of experience in nerve blocks.

## Ultrasound-guided ESPB or PVB

Upon entering the operating room, an 18-gauge intravenous (IV) cannula was used to establish an IV line in the forearm. Basal fluids were maintained using lactated Ringer's solution (RL) at a rate of 2–4 ml/kg/h. Vasoactive medications such as atropine, ephedrine, or epinephrine were readily available at the bedside for rescue purposes. Following the administration of 5–10 μg sufentanil intravenously, all patients were positioned in the lateral supine position to undergo regional block.

Ultrasound-guided ESPB: A 2–5 MHz convex array transducer (HFL38xi, Fujifilm SonoSite, Bothell, WA, USA) was used to guide the procedure. Starting from the C7 spinous process, the transducer was slid down until the T8 spinous process was identified. Subsequently, the transducer was placed longitudinally 2–3 cm lateral to the T8 transverse process. The trapezius and erector spinae muscles were visualized, and a 24-gauge needle was inserted from the caudal side to the cephalic side using an in-plane technique. The needle tip was confirmed to be positioned on the deep surface of the erector spinae muscle and the transverse process. Hydrodissection was then performed in the interfascial plane using three mL saline solution, followed by an injection of 20 mL of 0.33% ropivacaine. The same procedure was repeated for the opposite side of ESPB.

Ultrasound-guided PVB: The T8 transverse process was identified using the same ultrasound-guided technique as ESPB. After locating the pleura, a 24-gauge needle was inserted out-of-plane and directed towards the T8 transverse process. Once the needle contacted the transverse process, it was moved off the bone in a caudal direction to penetrate the superior costotransverse ligament. Aspiration was performed to confirm the absence of blood vessels or pleural injection. Following this confirmation, the local anesthetic was delivered, and the pleura was pushed down. The same procedure was performed for the opposite side PVB, with the same amount of ropivacaine administered through the T8 paravertebral spaces.

## Perioperative management

According to the patient's lean body weight (LBW) (*Ingrande, Brodsky & Lemmens, 2011*), propofol (1.5−2.5 mg/kg) and sufentanil (0.5 ug/kg) were intravenously injected for anesthesia induction. Rocuronium 0.6–1.2 mg/kg was also administered. Mechanical ventilation commenced following tracheal intubation. Anesthesia was maintained using propofol (4–8 mg kg$^{-1}$ h$^{-1}$), sevoflurane (1–2%), and remifentanil (0.05–0.15 ug kg$^{-1}$ min$^{-1}$). Rocuronium was intermittently injected. The Bispectral Index (BIS) value was kept between 40 and 60. Intraoperative blood pressure fluctuations were maintained within ± 20% of the baseline level.

After anesthesia induction, flurbiprofen (100 mg) was administered intravenously. At the end of the surgery, sugammadex (Bridion, Merck, Rahway, NJ, USA) was intravenously injected at a dose of 2 mg/kg of LBW. Once the patient met extubation criteria (*e.g.*, stable vital signs, adequate respiratory effort, and full consciousness), the tracheal catheter was removed and extubation was performed. Patients were then transferred to the PACU, where they were assessed by the anesthetist and subsequently discharged back to the ward by the nurse anesthetist once they met the criteria for discharge.

For postoperative analgesia, all patients received patient-controlled intravenous analgesia (PCIA) using a pump (REHN11, Renxian Medical Corporation, Jiangsu, China) with the following protocol: flurbiprofen 200 mg plus oxycodone 20 mg, diluted to 100 mL with normal saline. The pump parameters were set as follows: background free infusion dose, single dose of 5 mL, interval of 5 min, and limited dose of 20 mL.

## Study outcomes and measurements

The primary outcome was the QoR-15 score 24 h after the operation. The QoR-15 assesses 15 subjective factors, such as physical comfort (5 items), emotional state (4 items), physical independence (2 items), psychological support (2 items), and pain (2 items) (*Kleif et al., 2018*). Each item is scored from 0 to 10, resulting in a total score ranging from 0 to 150. A higher total score indicates a better quality of recovery. Secondary outcomes included the QoR-15 score at 48 h after the operation, Numeric Rating Scale (NRS) scores at rest and during coughing at 0.5, 2, 6, 12, 24, 36, and 48 h postoperatively, time to first ambulation, time to first anal exhaust, postoperative cumulative oxycodone consumption, the incidence of postoperative nausea and vomiting (PONV), as well as the complications associated with two types of nerve blocks, mainly including pneumothorax, local anesthetic toxicity, bleeding, nerve injury, and puncture site infection. Adverse events associated with the nerve block were noted during the initial phase of the nerve block procedure, whereas subsequent assessments of other outcomes were conducted postoperatively following post-anesthesia recovery.

## Statistical analysis

The sample size was determined using Gpower 3.1. Our primary outcome was the QoR-15 score at 24 h postoperatively, with the minimum clinically important difference for the QoR-15 score being 8 (*Myles et al., 2016*). Based on a pilot study with a sample size of 20, the QoR-15 score at 24 h after preoperative PVB at T8 was $122 \pm 12.1$ for patients undergoing LSG, while the QoR-15 score at 24 h after preoperative ESPB at T8 was $129 \pm 10.8$. Assuming $\alpha = 0.05$, $1 - \beta = 0.8$, a two-tailed test was conducted, resulting in a calculated sample size of 44 for each group. To account for a potential 20% dropout rate, a total of 110 participants were enrolled in this study.

R software (https://www.r-project.org/) was utilized to analyze all statistical data. In this study, age, BMI, duration of surgery, and cumulative oxycodone consumption postoperatively were treated as continuous variables following a normal distribution, expressed as mean $\pm$ standard deviation, and compared using independent samples $t$-test. QoR-15 scores, NRS scores, time to first ambulation, and time to first anal exhaust, as non-normally distributed continuous variables, were presented using the median (interquartile range). QoR-15 scores, time to first ambulation, and time to first anal exhaust was compared using the Mann–Whitney $U$ test, while NRS scores were compared using repeated measures analysis of variance. Gender, ASA physical status, and the incidence of PONV were considered categorical variables, presented as a number of cases (%), and compared using the chi-squared test or Fisher's exact test. A $P$-value $< 0.05$ was statistically significant.

## RESULTS

In this study, a total of 120 patients underwent LSG evaluation. Three patients were excluded due to not meeting the inclusion criteria, and five patients refused to participate. Consequently, 112 patients were randomly allocated to two groups. In the ESPB and PVB groups, two and three patients were respectively excluded due to mid-withdrawal. Ultimately, 107 patients successfully completed the study, with 54 in the ESPB group and 53 in the PVB group. The study flowchart is displayed in Fig. 1. There were no significant differences in patient demographics and operative characteristics between the two groups (Table 1).

There was no significant difference in the 24-hour QoR-15 score (median 131, IQR 112–140 *vs.* median 124, IQR 111–142.5; $P = 0.525$) and the 48-hour QoR-15 score (median 139.5, IQR 127–148 *vs.* median 141, IQR 121.5–148; $P = 0.908$) between the two groups (Fig. 2).

The postoperative NRS pain scores are displayed in Table 2. There were no significant differences in NRS pain scores at rest and during coughing between the two groups at 0.5, 2, 6, 12, 24, 36, and 48 h (all $P > 0.05$).

There were no significant differences in the postoperative time to first ambulation and time to first anal exhaust between the two groups ($P = 0.117$ and $P = 0.400$, respectively). Additionally, there were no significant differences in postoperative cumulative oxycodone consumption and the incidence of PONV between the two groups ($P = 0.168$ and $P = 0.868$, respectively) (Table 3).

Both groups did not experience any complications related to the two types of nerve block.

## DISCUSSION

In this trial, we discovered that patients with LSG could achieve a comparable level of postoperative recovery with a bilateral ultrasound-guided ESPB at T8, as opposed to a bilateral ultrasound-guided PVB at T8 prior to surgery. In addition, NRS pain scores within 48 h postoperatively, time to first ambulation, time to first anal exhaust, incidence of PONV, and incidence of nerve block complications were similar.

For decades, epidural analgesia has been considered the standard of care for pain management after thoracic and abdominal surgery (*Manion & Brennan, 2011*). However, several limitations have constrained its clinical application. Firstly, epidural analgesia may lead to hypotension and impaired mobility (*Hasselager, Hallas & Gögenur, 2022*). Second, puncturing failure rates of up to 32% highlight the need for improved placement verification methods beyond current limitations (*Hermanides et al., 2012*; *Motamed et al., 2006*). Thirdly, epidural puncture may result in site infections and nerve damage (*Jin et al., 2015*). Lastly, it imposes stringent requirements on blood coagulation function (*Mao et al., 2018*).

PVB involves the injection of local anesthetics into the wedge-shaped area on either side of the spine, with the solution diffusing widely along the vertebral bodies, intercostal spaces, and the epidural space, resulting in sensory, motor, and sympathetic nerve blockade,
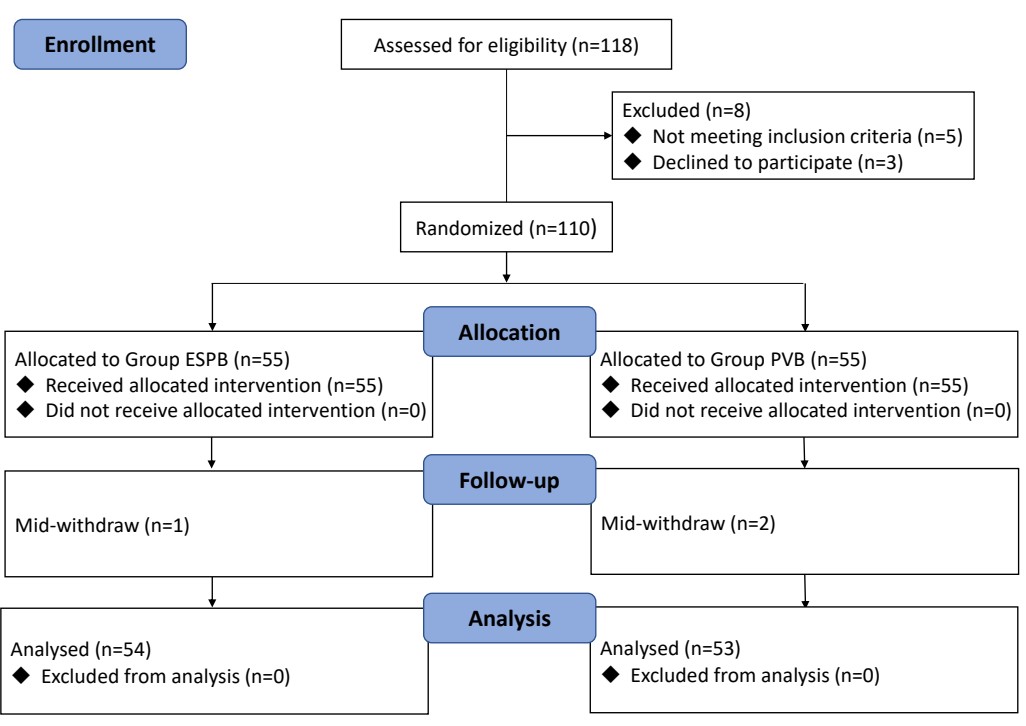

**Figure 1  CONSORT diagram of study.**

**Table 1  Patient demographics and operation characteristics.**

| Variables | Group ESPB ($n = 54$) | Group PVB ($n = 53$) | P value |
|---|---|---|---|
| Age, years | $31.6 \pm 6.4$ | $32.5 \pm 5.8$ | 0.453 |
| BMI, kg/m$^2$ | $35.9 \pm 2.4$ | $35.8 \pm 2.5$ | 0.774 |
| Gender, n (%) | | | 0.527 |
| Male | 14 (25.9) | 11 (20.8) | |
| Female | 40 (74.1) | 42 (79.2) | |
| ASA physical status, n (%) | | | 0.742 |
| II | 50 (92.6) | 48 (90.6) | |
| III | 4 (7.4) | 5 (9.4) | |
| Duration of surgery, min | $129.8 \pm 20.5$ | $128.2 \pm 18.2$ | 0.673 |
| Preoperative global QoR-15 score | 138.5 (131–144) | 141 (131.5–146) | 0.317 |

**Notes.**
Data are expressed as mean $\pm$ SD, numbers (%) or median (interquartile range).
Abbreviations: Group ESPB, Group erector spinae plane block; Group PVB, Group paravertebral block; BMI, Body mass index; ASA, American Society of Anesthesiologists; QoR, quality of recovery.

thus achieving analgesic effects. Numerous studies have illustrated that PVB can yield an analgesic effect comparable to epidural analgesia, accompanied by a reduced incidence of complications, positioning it as a viable alternative to epidural analgesia (*Davies, Myles & Graham, 2006*; *Joshi et al., 2008*). While PVB still presents potential complications, such as pneumothorax, nerve injury, and hypotension, and necessitates the anesthesiologist's

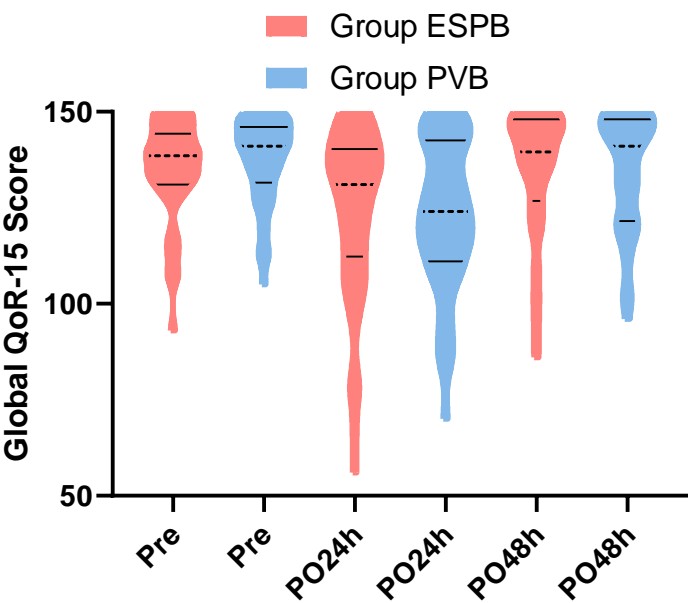

**Figure 2** **Distribution of QoR-15 scores at preoperative, 24-hour postoperative, and 48-hour postoperative time points in ESPB and PVB groups.** Violin plots depict probability density estimates using kernel density estimation. Median values are indicated by black dotted lines, and interquartile ranges by black solid lines. Group ESPB = Group erector spinae plane block, Group PVB = Group paravertebral block, Pre = Preoperative, PO24 h = Postoperative 24 h, PO48 h = Postoperative 48 h.

proficiency in ultrasound guidance, neuroanatomy, and precise positioning, it remains a procedurally challenging technique with an inherent risk of puncture failure (*Onishi et al., 2019*; *Şalvız et al., 2023*).

ESPB introduces an innovative approach to trunk nerve blockade, entailing the administration of local anesthetic into the myofascial plane between the erector spinae muscle and the transverse process of the vertebra. This action effectively obstructs the dorsal and ventral rami of the thoracolumbar nerves, achieving a multi-dermatomal sensory blockade across the anterior, posterior, and lateral chest, as well as the abdominal walls. Successful application of ESPB in treating thoracic neuropathic pain and achieving favorable outcomes has been reported in the literature (*Forero et al., 2016*). In contrast to PVB, ESPB targets a plane distant from the pleura and major neurovascular structures with easily identifiable ultrasound-based anatomical landmarks (*Smith, Barrington & St Vincent's Hospital, Melbourne, 2020*). ESPB presents a lower risk of systemic local anesthetic toxicity and pleural puncture, exerting minimal impact on intraoperative blood pressure, and requiring relatively lower coagulation function (*Aydın et al., 2019*; *Huang & Liu, 2020*). Consequently, ESPB is regarded as a comparatively straightforward and safe regional blockade. Moreover, studies indicate that proficiency in ESPB can be acquired more rapidly, and the procedural time is shorter compared to PVB (*Moustafa et al., 2020*). Therefore, in the context of postoperative analgesia for thoracic and abdominal surgery, ESPB emerges as a preferable alternative to PVB.

**Table 2 Postoperative NRS pain scores.**

|  | Group ESPB ($n = 54$) | Group PVB ($n = 53$) | *P* value |
|---|---|---|---|
| NRS at rest |  |  |  |
| 0.5 h | 0 (0–2) | 1 (0–2) | 0.589 |
| 2 h | 2 (1–2) | 2 (2–3) | 0.234 |
| 6 h | 2 (1–2) | 2 (2–3) | 0.441 |
| 12 h | 2 (1–2) | 2 (1–3) | 0.765 |
| 24 h | 1 (1–2) | 2 (1–2) | 0.763 |
| 36 h | 1 (1–1) | 1 (1–2) | 0.103 |
| 48 h | 1 (0–1) | 1 (1–2) | 0.079 |
| NRS during coughing |  |  |  |
| 0.5 h | 1 (0–2) | 1 (0–3) | 0.501 |
| 2 h | 1 (1–2) | 2 (1–3) | 0.333 |
| 6 h | 2 (1–2) | 2 (2–3) | 0.172 |
| 12 h | 2 (1–2) | 2 (1–2) | 0.579 |
| 24 h | 1 (1–2) | 2 (1–2) | 0.208 |
| 36 h | 1 (0–1) | 1 (1–2) | 0.154 |
| 48 h | 1 (0–1) | 1 (1–1) | 0.051 |

Notes.
Data are expressed as median (interquartile range).
Abbreviations: NRS, numerical rating scale; Group ESPB, Group erector spinae plane block; Group PVB, Group paravertebral block.

**Table 3 Secondary outcomes during the study period.**

| Variables | Group ESPB ($n = 54$) | Group PVB ($n = 53$) | *P* value |
|---|---|---|---|
| Time to first ambulation, h | 2 (2–2) | 2 (2–2) | 0.117 |
| Time to first anal exhaust, h | 23 (18–26) | 21 (18–25) | 0.400 |
| Cumulative oxycodone consumption, mg | 19.8 ± 11.2 | 17.0 ± 9.6 | 0.168 |
| PONV, n (%) | 14 (25.93%) | 13 (24.53%) | 0.868 |

Notes.
Data are expressed as mean ± SD, median (interquartile range) or numbers (%).
Abbreviations: Group ESPB, Group erector spinae plane block; Group PVB, Group paravertebral block; PONV, postoperative nausea and vomiting.

In obese patients undergoing ultrasound-guided PVB and ESPB, the presence of increased subcutaneous fat poses challenges. This includes augmented depth at the targeted puncture site, heightened difficulty in visualizing the needle, prolonged operation times, and the potential restriction of local anesthetic diffusion to nerve roots (*Butcher et al., 2014*; *Kula, Riess & Ellinas, 2017*). Despite accumulating clinical experience in ultrasound-guided nerve blocks, managing patients with a BMI >30 kg/m$^2$ remains a persistent challenge (*Franco et al., 2006*). Presently, there is limited research on the utilization of PVB and ESPB in obese individuals undergoing bariatric surgery, with scarce evidence regarding their efficacy and safety. Additionally, it remains unclear whether ESPB can adequately replace PVB in obese patients. Consequently, this study was conducted to compare the impact of these two nerve blocks on postoperative recovery quality. We sought to provide clinical

evidence regarding the safety and effectiveness of both nerve blocks in obese patients and to determine whether ESPB can serve as a viable alternative to PVB.

The QoR-15 score serves as a valuable tool for evaluating the quality of postoperative recovery in patients. Derived from the QoR-40 score it offers a simpler and more convenient application (*Kleif et al., 2018*). Recent research suggests that both scoring systems exhibit similar psychometric properties, thereby enhancing the broader clinical applicability of the QoR-15 score (*Myles et al., 2022*). Previous studies have predominantly focused on patients' clinical outcomes, overlooking comprehensive evaluations of patient recovery. In this study, we selected the QoR-15 score as the primary outcome, aligning with the current patient-centered philosophy. The observed difference in QoR-15 scores at 24 and 48 h postoperatively between the two groups in this study did not reach statistical significance, indicating that both types of nerve blocks result in comparable postoperative recovery quality. According to *Myles et al. (2016)* a QoR-15 score of at least 118 signifies good postoperative recovery. In our study, the median QoR-15 scores for both groups at 24 and 48 h postoperatively surpassed 118, indicating favorable postoperative recovery quality with both types of nerve blocks.

Following bariatric surgery, the prevalence of moderate to severe postoperative pain can reach up to 65%, primarily attributed to abdominal wall and visceral pain (*Yurttas et al., 2023*). PCIA alone proves inadequate in addressing this issue, emphasizing the essential role of nerve blockade within multimodal analgesia strategies. Within the initial 48 h postoperatively, both groups demonstrated median NRS pain scores at rest and during coughing of less than 3, signifying effective analgesia from both types of nerve blocks. This contributed to a reduction in the incidence of moderate to severe postoperative pain in the context of bariatric surgery. Additionally, the time to first ambulation and anal exhaust in both groups showed numerical decreases compared to LSG patients without preoperative nerve block at our institution (where the median time to first ambulation and anal exhaust without nerve block were 9 h and 31 h, respectively). Effective analgesia not only alleviates pain but also facilitates early mobilization, promoting the recovery of gastrointestinal function, and ultimately leading to a shortened time to first anal exhaust.

Patients undergoing bariatric surgery face a heightened susceptibility to PONV, as evidenced by *Halliday et al. (2017)* who reported a 65% incidence of PONV after LSG. In contrast, our study demonstrated a significantly lower incidence of approximately 25%. This reduction in PONV incidence is likely attributed to the preoperative nerve blockade, which diminishes the need for opioid medications during and after surgery. The favorable analgesic effects of the nerve blockade further contribute to early patient mobilization and the recovery of gastrointestinal function.

Notably, no instances of complications related to the two types of nerve blocks were observed in this study, providing additional evidence for the safety of employing these techniques in obese patients. Furthermore, *Şalvız et al. (2023)* revealed that PVB led to a reduction in intraoperative blood pressure, whereas ESPB had no such effect. This discrepancy may be linked to PVB causing dural spread and direct suppression of sympathetic nerves.

This study has several notable limitations that should be acknowledged. Firstly, the BMI range of the included obese patients was confined to 30–40 kg/m$^2$. Future investigations are warranted to assess the safety and effectiveness of the two nerve blocks in morbidly obese patients with a BMI exceeding 40 kg/m$^2$. Secondly, being a single-center study, the findings may benefit from validation through multicenter studies to enhance the generalizability of these nerve blocks. Variations in population demographics and anesthesiologists' practices across different centers could impact the study outcomes. Thirdly, the absence of a group without nerve block in this study was deliberate, as nerve blocks are known to reduce opioid consumption and alleviate postoperative pain. However, this decision prompts inquiry into whether nerve blocks also contribute to an improvement in the quality of postoperative recovery for patients. According to our unpublished research, the 24-hour QoR-15 score for patients without nerve block was $108 \pm 7.2$, whereas the QoR-15 scores for both groups of patients in this study exhibited an increase. Fourthly, the pain and temperature perception level of patients was not tested after completing the nerve block. Finally, the time of the patients' first analgesic request was not recorded, precluding an assessment of the duration of action of the two nerve blocks.

## CONCLUSIONS

In patients undergoing LSG, preoperative bilateral ultrasound-guided ESPB demonstrated a comparable postoperative recovery quality and NRS pain scores within the initial 48 h when compared to preoperative bilateral ultrasound-guided PVB. Both patient groups exhibited similar durations for the time to first ambulation, time to first anal exhaust, and incidence of PONV. Notably, no instances of nerve block-related complications were observed in either group. These findings reinforce the safety and efficacy of both nerve blocks in obese patients undergoing bariatric surgery, lending support to the potential of ESPB as a viable substitute for PVB.

## ACKNOWLEDGEMENTS

We express our gratitude to the bariatric surgery department for their support in facilitating our patient recruitment process.

### Funding

The authors received no funding for this work.

### Competing Interests

The authors declare there are no competing interests.

### Author Contributions

- Guanyu Yang conceived and designed the experiments, analyzed the data, prepared figures and/or tables, authored or reviewed drafts of the article, and approved the final draft.

- Pengfei Wang conceived and designed the experiments, performed the experiments, authored or reviewed drafts of the article, and approved the final draft.
- Yue Yin conceived and designed the experiments, performed the experiments, authored or reviewed drafts of the article, and approved the final draft.
- Huan Qu conceived and designed the experiments, performed the experiments, authored or reviewed drafts of the article, and approved the final draft.
- Xin Zhao conceived and designed the experiments, performed the experiments, authored or reviewed drafts of the article, and approved the final draft.
- Xiaogao Jin conceived and designed the experiments, analyzed the data, prepared figures and/or tables, authored or reviewed drafts of the article, and approved the final draft.
- Qinjun Chu conceived and designed the experiments, analyzed the data, prepared figures and/or tables, authored or reviewed drafts of the article, and approved the final draft.

## Human Ethics

The following information was supplied relating to ethical approvals (i.e., approving body and any reference numbers):

The Medical Ethics Committee of Zhengzhou Central Hospital (ethical number: 202109).

## Clinical Trial Ethics

The following information was supplied relating to ethical approvals (i.e., approving body and any reference numbers):

the Chinese Clinical Trial Registry.

## Data Availability

The raw measurements are available in the Supplementary File.

## Clinical Trial Registration

The following information was supplied regarding Clinical Trial registration:

ChiCTR2100042846.

## Supplemental Information

Supplemental information for this article can be found online at http://dx.doi.org/10.7717/peerj.17431#supplemental-information.

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
