# Peer review of "Erector spinae plane block versus paravertebral block on postoperative quality of recovery in obese patients undergoing laparoscopic sleeve gastrectomy: a randomized controlled trial"

_PeerJ, doi:10.7717/peerj.17431_

## Round 0.1 · original submission · Major Revisions

The reviewers have provided many important issues. Thank their contribution.

**Language Note:** The review process has identified that the English language must be improved. PeerJ can provide language editing services - please contact us at [email protected] for pricing (be sure to provide your manuscript number and title). Alternatively, you should make your own arrangements to improve the language quality and provide details in your response letter. – PeerJ Staff

·

Basic reporting

Many thanks for asking me to review this manuscript
To make this manuscript more factual and clinically relevant, I would suggest the following revisions
Grammatical and linguistic errors: throughout the manuscript, please amend them.

Experimental design

1-Please define adverse effects.
2-would you please specify which test was used for which variables rather than make generalized statements in your description of the statistical analysis

Validity of the findings

Include more detail regarding the time to event measures. Specify the time in which the observation started. State as well that median time to analgesic request was estimated, and the 95% CI

Additional comments

Need English editing

Reviewer 2 ·

Basic reporting

Reviewing Manuscript
‘’Erector spinae plane block versus paravertebral block on postoperative quality of recovery in obese patients undergoing laparoscopic sleeve gastrectomy: a randomized controlled trial’’
I read the article with interest. The hypothesis and abstract established in the article are quite interesting. However, the literature review is insufficient. The introduction and discussion for the Erector spinae plane block (ESPB), which is the most interesting facial plane block of recent years, is quite weak.

Experimental design

No comment

Validity of the findings

I read the article with interest. The hypothesis and abstract established in the article are quite interesting. However, the literature review is insufficient. The introduction and discussion for the Erector spinae plane block (ESPB), which is the most interesting facial plane block of recent years, is quite weak.

Reviewer 3 ·

Basic reporting

Without a doubt, an interesting clinical trial that describes the differences between 2 regional analgesia techniques that are not commonly used in bariatric procedures.

It would be interesting if the authors commented on what their postoperative pain management protocol was like before this study and if they modified it with respect to its results.

After reviewing the content, I must make some editorial comments:
-In the material and methods section, it is not clear what BMI range they use as a criterion for including patients in the study.

Experimental design

The methodological design of the study is clearly stated.

Validity of the findings

It is not very clear to me the contribution to the postoperative management of patients undergoing gastric sleeve, I think that the authors should address more in the discussion that the results of this study contribute to daily clinical practice.

Additional comments

No.

·

Basic reporting

The research is a modest contribution to ongoing discussion about the use of regional analgesia for bariatric surgery. I just need clarification of some points
Page 2 line 40:
I prefer to remove the word "developed" as obesity is a common problem in developed and developing countries.
Page 5 line 166:
You mentioned "In this randomized controlled trial" in the discussion and it was mentioned previously in the methods section so it is better to be removed
In the exclusion criteris you mentioned "changed surgical procedures". It is not precise expression and it is better to add (patients scheduled for LSG) in the inclusion criteria.
Please wright the QoR-15 score in a table for better understanding
You should wright a summary of results in first paragraph of discussion section not only the primary outcome.
In conclusion you should mention other outcomes
Discussion is very short more details should be added

Experimental design

Why is BMI limited to 30-40 kg/m2?
Why did you perform the regional block in lateral position? it is easy to be performed in sitting position especially in obese patients.
Please mention the formula used to calculate LBW.
You mentioned " Intraoperative blood pressure fluctuations were maintained within ± 20% of the baseline level" what about patients with blood pressure fluctuation more or less than 20% of the baseline level?
What do you mean by " Tracheal extubation was performed if indicated"
Page 5 line 182:
You mentioned "ultrasound images of ESPB were easier to interpret than those of PVB" It is a subjective opinion and may be not the case for other anesthetists so it is better to talk only about details of sono-anatomy .
Page 6 line 213:
You mentioned "According to our previous research" specify which research and add a reference

Validity of the findings

Why did you not measure length of PACU stay??
Please mention the details of PCA use (concentration, infusion rate, max limit, bolus dose) and what did you give during breakthrough pain?????
You mentioned "A higher total score indicates a better quality of recovery" it is not precise it is better to mention numeric limit.
Are there any side effects other than PONV? Please clarify
You mentioned" Sample size calculation Based on our preliminary study" was it a pilot study??? What were the details of this study e.g. sample size, primary outcome

Page 5,6 lines 195-198:
It is not suitable to compare the results of the study with the results of your hospital in general especially you mentioned "numerically" with no numbers or statistics added
When did you start PCA? if it was started immediately postoperative why did you include NRS in the measurements? PCA controlled the pain allover the time and it was not related to the block and if PCA was started after the analgesic request from the patients you should include the time to first analgesic request which indicate the duration of analgesia provided by the block

---

## Round 0.2 · Minor Revisions

There are some minor questions from Reviewer 5 to be addressed.

·

Basic reporting

Thanks for your great response

Experimental design

Thanks for your great response

Validity of the findings

Thanks for your great response

Reviewer 5 ·

Basic reporting

No comment.

Experimental design

Firstly, what is the design of this clinical trial, a non-inferiority study or an equivalence study? What is the hypothesis of this study? That is why the sample size calculation is confusing. The authors presented QOR-15 after both regional blocks from a pilot study, but the authors did not present their hypothesis.
Secondly, did the authors test the block levels after ESPB and PVB? How did they know that the blocks were effective.

Validity of the findings

Did the patients have any pain rescue besides prescribed PCA with flurbiprofen and oxycodone (20mg)? Telling from table 3, nearly all oxycodone in PCA were consumed (19.8 mg vs. 17.0mg)?

---

## Round 0.3 · Minor Revisions

Please address the remaining comments of Reviewer 5

Reviewer 2 ·

Basic reporting

I evaluated the article as successful.

Experimental design

I evaluated the article as successful.

Validity of the findings

None

Additional comments

I congratulate the authors for their contribution to their field.

·

Basic reporting

No comment

Experimental design

No comment

Validity of the findings

No comment

Additional comments

No comment

Reviewer 5 ·

Basic reporting

No comment.

Experimental design

Design of an equivalence study is not well described. Block levels could also be tested after surgery.

Validity of the findings

No comments.

Additional comments

Line 92 Page 3, how to place a patient in a lateral supine position? Could the authors provide photos about how to perform ESPB and PVB on both sides of the patients?

The authors could consult a statistical specialist about how to design an equivalence study. Usually an equivalent margin is set. When no statistically significance is detected between two groups, it could not be concluded that two interventions are equivalent in reducing postoperative pain.

---

## Round 0.4 · accepted · Accept

It is a good revision, and could be accepted.